# Social and Economic Factors and Malnutrition or the Risk of Malnutrition in the Elderly: A Systematic Review and Meta-Analysis of Observational Studies

**DOI:** 10.3390/nu12030737

**Published:** 2020-03-11

**Authors:** Maria Besora-Moreno, Elisabet Llauradó, Lucia Tarro, Rosa Solà

**Affiliations:** 1Universitat Rovira i Virgili, Facultat de Medicina i Ciències de la Salut, Functional Nutrition, Oxidation, and Cardiovascular Diseases Group (NFOC-Salut), 43201 Reus, Spain; mariadelaserra.besora@urv.cat (M.B.-M.); lucia.tarro@urv.cat (L.T.); rosa.sola@urv.cat (R.S.); 2Centre Tecnològic de Catalunya, Unitat de Nutrició i Salut, Eurecat, 43204 Reus, Spain; 3Hospital Universitari Sant Joan de Reus, Department of Internal Medicine, 43204 Reus, Spain

**Keywords:** malnutrition, nutritional status, elderly, socioeconomic factors

## Abstract

Malnutrition in the elderly could be tackled by addressing socioeconomic factors. This study aimed to determine the magnitude of the relationship between socioeconomic factors and the malnutrition or malnutrition risk (MR) in the elderly. The PubMed and SCOPUS databases were searched for observational studies that included assessment of malnutrition or/and MR and socioeconomic variables (educational level, living alone, marital status, income and occupational level, feeling of loneliness, place of residence, and food expenditure) in ≥60-year-old subjects, published in English among 2000–2018 (PROSPERO: CRD42019137097). The systematic review included 40 observational studies (34 cross-sectional and 4 cohort studies) and 16 cross-sectional studies in the meta-analysis (34,703 individuals) of malnutrition and MR in relation to low educational level (Odds Ratio (OR): 1.48; 95% Confidence Interval (CI): 1.33–1.64; *p* < 0.001), living alone (OR: 1.92; 95% CI: 1.73–2.14; *p* < 0.001), being single, widowed, or divorced (OR: 1.73; 95% CI: 1.57–1.90; *p* < 0.001), and low income level (OR: 2.69; 95% CI: 2.35–3.08; *p* < 0.001), and considering these four socioeconomic factors, malnutrition and MR is associated with them (OR: 1.83; 95% CI: 1.73–1.93; *p* < 0.001). Malnutrition and MR could be reduced by increasing economic level, supporting people living alone or being single, widowed, and divorced, and improving lifelong learning.

## 1. Introduction

Nowadays, the aging of the population is a consequence of the increase in life expectancy [1]. In Europe, between 2015 and 2050, the proportion of the world’s population over 60 years will nearly double from 12% to 22%, with a different distribution of gender and by country [2].

Aging can be associated with malnutrition, which is a public health problem characterized by a multifactorial physiological state [3,4]. Malnutrition is defined as an insufficient nutritional intake or absorption, which leads to a decrease in fat and muscle mass [5,6]. From the European Society for Clinical Nutrition and Metabolism (ESPEN) definition, malnutrition is diagnosed by a body mass index (BMI) of <18.5 kg/m^2^ or by meeting two of these three criteria: unintentional weight loss (>10% in an indefinite time period or >5% over the last three months) combined with either a low BMI (BMI of <20 kg/m^2^ if <70 years of age, or <22 kg/m^2^ if ≥70 years of age) or a low fat-free mass index (FFMI) score (FFMI of <15 and <17 kg/m^2^ in women and men, respectively) [7]. The overall prevalence of malnutrition in the elderly ranges from 1% to 24.6% [8]. In addition, 50% of the elderly in rehabilitation, 20% in residential care, and 40% in hospitals are malnourished [9]. As a result of population aging, the malnutrition prevalence is increasing, which is expected to reach 29.1% by 2080 [1].

However, before reaching the need of a malnutrition diagnosis, the use of validated tools could be used to determine malnutrition risk [6,7]. According to the ESPEN, the Mini Nutritional Assessment (MNA) is the most effective tool for screening and evaluating the risk of malnutrition in the elderly [6,10,11], composed of four areas related to anthropometry, clinical status (illness, medications, psychological stress, neuropsychological problems), dietary assessment, and self-perception about health and nutrition [10]. Other tools used to assess the risk of malnutrition include the Malnutrition Screening Tool (MUST) and the Nutrition Risk Screening (NRS), but these do not include the evaluation of important malnutrition risk factors such as functional, psychological, and cognitive parameters [10].

Consequently, a state of malnutrition can cause an impairment of quality of life, especially in the elderly, and can lead to increased healthcare costs and hospital stays [12].

Additionally, malnutrition is related to health, environmental, and social factors or determinants [6]. The elderly are more vulnerable to developing a worse nutritional status because of their lack of teeth, their loss of taste, or their poor mobility [5]; as a consequence, these factors can affect dietary intake. The social factors associated with malnutrition include lifestyle, loneliness, isolation, marital status, educational level, socioeconomic level, and place of residence [3,6]. Those that are single, widowed, or divorced are the most malnourished or at the most risk of malnutrition, especially the widowed [13]. The elderly with low educational levels have worse nutritional status due to the lack of cooking skills or insufficient knowledge to make healthy food choices [14]. Furthermore, there is a higher prevalence of malnutrition in people who live in rural areas [8]. Another determinant of malnutrition is income level, with a strong relationship between the levels of poverty and malnutrition in the elderly [13]. Thus, the social and economic determinants of malnutrition risk and/or malnutrition development should be analyzed to establish which determinants affect malnutrition and in which magnitude, as a basis for taking possible actions to reduce malnutrition in the elderly.

Thus, the hypothesis was that social and economic factors—such as low educational level, living alone, being single, widowed, or divorced, low income level, low previous (before 60 years old) or current occupational level, feeling of loneliness, living in rural areas, and low food expenditure—are related to malnutrition and malnutrition risk in the elderly.

The main objective of the present systematic review and meta-analysis was to assess and determine the magnitude of the relationship between socioeconomic factors and the malnutrition or malnutrition risk in the elderly ≥60 years old.

## 2. Materials and Methods

We performed a systematic review and meta-analysis of observational studies regarding malnutrition and malnutrition risk and their relation to socioeconomic factors in the elderly. This study was reported in accordance with the Preferred Reporting Items for Systematic Reviews and Meta-Analyses (PRISMA) (Appendix A) and it was registered in PROSPERO International Prospective Register of Systematic Reviews (CRD42019137097).

### 2.1. Search Strategy

An exhaustive literature search in the PubMed and SCOPUS electronic databases was carried out, as these databases are considered the largest and the most multidisciplinary, covering medical and social sciences journals that include the principal outcomes of the present meta-analysis (malnutrition and socioeconomic factors). Depending on the study, socioeconomic factors were assessed separately as social factors and economic factors. Regarding nutritional status, some studies assessed the malnutrition and the malnutrition risk separately, while other studies evaluated them together. For this reason, the search was done using the following keywords combined with each other: malnutrition, nutritional status, older, elderly, social factors, social, economic, and socioeconomic. The search strategies in PubMed were: “older” or “elderly” and “social” and “malnutrition”, “older” or “elderly” and “economic” and “malnutrition”, “nutritional status” and “social factors”; and in SCOPUS were: “economic” and “malnutrition” and “older” or “elderly”, “social” and “malnutrition” and “older” or “elderly”.

### 2.2. Study Selection

The inclusion criteria were observational studies (case-control, prospective cohort, and cross-sectional studies), with the target population ≥60 years old, in which publications with data on malnutrition status, as well as social and economic variables, were published in English between January 2000 and December 2018.

The studies that did not meet all of the inclusion criteria were excluded. Additionally, studies that only included populations with illness or comorbidities were also excluded in order to focus on the general population, who was not selected taking into account a specific disease, to be able to generalize the results.

### 2.3. Article Review and Data Extraction

The search strategy of the studies that matched the inclusion criteria was carried out with the online platform Covidence (Covidence systematic review software, Veritas Health Innovation, Melbourne, Australia; available at www.covidence.org). Firstly, the studies were selected based on the title information. Subsequently, the abstracts of these selected studies were checked to see if they met or not the inclusion and exclusion criteria. Finally, those studies that met the inclusion criteria were full-text assessed. When the studies did not have all of the information necessary to be included in the present study, an email was sent to the authors of the publication requesting this information. The data extraction was carried out independently. Of the included studies, the following variables were collected: authors, year of study development, the country of implementation, the type of study, number of participants, target population’s age, gender, social factors (educational level, living alone or cohabit, marital status, feeling of loneliness, place of residence), economic factors (income level, occupational level, food expenditure), nutritional status screening tool, nutritional status (risk of malnutrition, malnutrition, and optimal nutritional status), and study quality. The previous or current occupational level was divided into low occupational levels (such as farmer, breeder, housewife, laborer, or hard manual or physical work) and high occupational levels (such as employee, businessman, worker, administration, or jobs involving higher intellectual effort).

### 2.4. Risk of Bias in Individual Studies

To facilitate the quality assessment of the present study and to minimize the risk of bias in the included studies, the Quality Assessment Tool for Observational Cohort and Cross-Sectional Studies [15] was used with the following criteria: the cross-sectional studies were evaluated on 10 points, because questions 6, 7, 12, and 13 are only for cohort studies, while the prospective cohort studies were evaluated on 14 points. Observational studies were divided into three quality categories (high, medium, and low) with their respective cut-off points: (a) for cross-sectional studies, quality categories were: low quality (≤3 points), medium quality (4–7 points), and high quality (8–10 points); and (b) for prospective cohort studies, quality categories were: low quality (≤5 points), medium quality (6–9 points), and high quality (10–14 points).

### 2.5. Statistical Analysis: Meta-Analysis

To carry out the meta-analysis, we used the Review Manager (RevMan) (computer program: version 5.3. Copenhagen: The Nordic Cochrane Centre, The Cochrane Collaboration, 2014). To select studies for the meta-analysis, it was taken into account whether they used the MNA as a nutritional evaluation tool (as this was the most used tool to evaluate the nutritional status in the elderly among the studies) and evaluated the socioeconomic variable with the same criteria. To include a study, information about the percentage or number of subjects with a risk of malnutrition or malnutrition-related to the socioeconomic variable was required. In contrast, studies that assessed all outcomes via multivariate analysis were excluded from the meta-analysis. Thus, we only performed the meta-analysis of each socioeconomic factor, in which at least 4 studies evaluated this factor and followed the inclusion criteria described previously. In addition, an analysis was performed in general (all variables in one) and by a subgroup of the same variables. To evaluate the heterogeneity of the studies, we used the I^2^ statistic. When the heterogeneity was over 85%, they were analyzed with randomized and non-fixed effects. A *p*-value of <0.05 was considered statistically significant.

## 3. Results

A total of 363 studies were obtained from the literature search, of which 60 were eliminated by duplicity. Of the remaining 303 studies, 197 were excluded according to the inclusion and exclusion criteria based on the information provided in the title and the abstract. A total of 106 studies were full-text assessed, and 66 were excluded because of the following reasons: results not related to the outcomes studied (*n* = 48), text not available (*n* = 8), non-representative population sample (*n* = 5), and wrong study design (*n* = 5). Finally, as shown in Figure 1, 40 studies were included in the systematic review [13,14,16,17,18,19,20,21,22,23,24,25,26,27,28,29,30,31,32,33,34,35,36,37,38,39,40,41,42,43,44,45,46,47,48,49,50,51,52,53] and 16 studies in the meta-analysis [13,20,21,25,26,29,30,35,36,39,42,43,45,51,52,53].

### 3.1. Characteristics of the Included Studies in the Systematic Review

Of the total 40 included studies, 36 were cross-sectional [13,14,17,18,20,21,22,23,24,25,26,27,28,29,30,31,32,34,35,36,37,38,39,40,41,42,43,44,45,46,47,48,49,51,52,53] and four were cohort studies [16,19,33,50]. The study population was 60 years old and above, except for two studies that focused only on people aged over either 85 years [26] or 90 years [37]. The total number of participants in the different studies ranged from 67 subjects [48] to 15,669 subjects [49], with a total of 61,818 subjects included in the present systematic review. All of the studies included people of both genders, except two studies [50,51]—one of them was only focused on men [50] and the other only on women [51]. Of the 40 studies included in the systematic review, according to the Development Human Index from the United Nations Development Program (UNDP) [54], 33 were conducted in regions with a high development human index (Poland [39], Sweden [38,40], China [30,37,53], Turkey [52], Canada [49,50], Italy [14,35], Taiwan [16], Lebanon [17,20,25], Brazil [18], Ireland [19,31], Finland [21], Spain [13,22,26], Norway [23,34], Portugal [24], Korea [27], Iran [28], Sri Lanka [29], New Zealand [48], Mexico [32], Amsterdam [33], Malaysia [36], and Shanghai [51]), five were conducted in regions with a medium development human index (Bangladesh [41,44], India [42], South Africa [43], and Egypt [45]), and two were conducted in regions with a low development human index (Central Africa [46] and Nigeria [47]).

Of the 40 observational studies included in the systematic review, regarding the tools used to assess the nutritional status, 29 studies used the MNA [13,14,16,17,18,20,21,23,24,25,26,28,29,30,31,32,34,35,36,37,39,40,41,42,43,45,51,52,53], one of which also used the Norwegian version of the Nutritional Form For the Elderly (NUFFE-NO) [34]. Of the other studies, five used only the Body Mass Index (BMI) [19,22,33,44,46] and one also used self-reported weight loss [19], four used the Senior in the community: risk evaluation for eating and nutrition (SCREEN II) [38,48,49,50] and two used the Nutritional Screening Initiative (NSI) [27,47]. Only three studies considered depression as a pathology added to malnutrition [18,28,43].

Of the total of 40 observational studies included in the systematic review, eight different socioeconomic variables were evaluated: 31 studies evaluated the educational level [13,14,16,18,19,20,22,24,25,26,27,28,29,30,33,35,36,37,39,41,42,43,44,45,46,47,48,49,51,52,53], 27 the fact of living alone or cohabiting [13,14,19,21,22,25,26,27,28,29,31,32,36,37,38,39,40,42,43,44,45,48,49,50,51,52,53], 24 the marital status [13,14,16,19,23,24,26,27,29,30,33,34,35,36,39,41,42,44,45,46,48,51,52,53], 20 the income level [14,16,20,22,24,25,27,29,30,33,35,37,39,41,42,43,44,45,47,51,52], 12 the occupational level [14,28,30,34,35,36,37,44,45,46,51,53], six the feeling of loneliness [13,17,21,33,34,36], three the place of residence [28,39,51], and one the food expenditure [41] (Table 1).

### 3.2. Educational Level and Malnutrition/Malnutrition Risk

Of the 31 studies that evaluated educational level, 15 established significant relationships with regard to malnutrition or malnutrition risk [14,22,25,27,28,35,39,41,42,45,46,47,49,52,53], two suggested a significant trend [30,44], and 14 did not present significant differences [13,16,18,19,20,24,26,29,33,36,37,43,48,51] (Appendix A). Of those 15 with signification, 11 studies showed a negative association, that is, the lower the educational level, the higher the malnutrition risk [14,25,28,35,39,41,42,45,46,49,53]. Five of these evaluated the Odds Ratio (OR) [35,42,45,46,49], revealing an increased malnutrition risk of between 1.3 and 8 times more in those who had low educational levels or fewer years of schooling [35,42,45,46,49]. In addition, one study presented results with the β coefficient [41], showing that the highest educational level was significantly associated with the greatest score in the MNA questionnaire [41]. Instead, the results of four studies were significantly contrary. On the one hand, two studies established that a low educational level protected against malnutrition [22,52]—one of which even related a low educational level with higher overweight and obesity risk [22]. On the other hand, one of the other studies indicated that the elderly with fewer years of schooling had better nutritional profiles [27], while the other study established a positive relationship between educational level and malnutrition according to the Pearson correlation [47].

### 3.3. Living Alone or Cohabiting and Malnutrition/Malnutrition Risk

A total of 27 studies evaluated the relationship between living alone or cohabiting and the malnutrition or malnutrition risk, of which 13 showed statistically significant results [13,14,21,27,28,32,38,39,45,49,50,51,53] and 14 showed no significance [19,22,25,26,29,31,36,37,40,42,43,44,48,52] (Appendix A). Of those 13 studies with statistically significant results, 11 showed that the elderly who lived alone had better malnutrition than the elderly who cohabited with other people—whether the spouse, sons, grandsons, or other family or friends [13,14,21,27,28,32,38,39,49,50,53]. Two of these evaluated the OR [32,38] and one of them also evaluated the β coefficient [32]. An increased risk of malnutrition was demonstrated in the elderly who lived alone [32,38], and it was observed that living alone increased malnutrition risk by 1.8 times [32]. In addition, the increased risk was found both in men and women who lived alone [38]. One study found that there was a higher significant percentage of people living with a spouse who had a better nutritional status [13]. However, two studies with significant results showed contrary results, establishing that of those with malnutrition, there was a higher percentage who lived together [45,51].

### 3.4. Marital Status and Malnutrition/Malnutrition Risk

A total of 24 studies evaluated the relation between the marital status and malnutrition or malnutrition risk, of which 11 showed significant results [14,19,27,30,33,34,35,39,42,45,53] and 13 were not significant [13,16,23,24,26,29,36,41,44,46,48,51,52] (Appendix A). Of those 11 studies with significant results, 10 showed that the elderly who were single, widowed, or divorced had a greater malnutrition risk than those who were married [14,19,27,33,34,35,39,42,45,53]. Only one study showed significance in relation to people with malnutrition who were in day centers, pointing out that a significantly higher percentage of these were widows [14]. A further four evaluated the OR [19,34,42,45] and one the Hazard Ratio (HR) [33]. All of the studies that evaluated the OR concluded that the risk of malnutrition was 1.64 times higher in single or divorced elderly [19], 2.19 times higher in single, divorced, or widowed people [42], 2.99 times higher in single elderly [34], and 29.4 times higher in divorced or widowed elderly [45]. HR also established that not having a partner was associated with a higher malnutrition risk [33]. Only one study obtained significant results in contrast to those defined above, expressing that of those elderly with a worse nutritional status, there was a higher percentage who were married [30].

### 3.5. Income Level and Malnutrition/Malnutrition Risk

Of the total of 20 studies that evaluated the income level, 13 established significant relationships [14,22,24,25,27,35,39,41,42,43,45,47,51], one suggested a significant trend [44], and six did not present significant differences [16,20,29,30,33,37] (Appendix A). Of those 13 studies with significant results, 12 suggested that a low income level was associated with a higher risk of malnutrition [14,22,24,25,27,35,39,41,42,43,45,51]. Of these 12 studies with a negative association, seven evaluated the OR [22,24,35,42,43,45,51] and three the β coefficient [41,43,51]. The OR of five studies determined that the malnutrition risk ranged between 1.31 [22] and 64.7 [45] times more in the elderly with low income levels. In another study, the OR showed that the elderly with low socioeconomic status had 6.01 times more risk of malnutrition [42]. One study concluded that having income and receiving finance on a regular basis is significantly associated with an increase in the score in the MNA questionnaire [41]. Lastly, it was found that a high economic level protects against malnutrition [51]. In contrast, however, from the significant studies, one study determined that at a high-income level there are higher levels of malnutrition [47].

### 3.6. Occupational Level and Malnutrition/Malnutrition Risk

Of the total of 12 studies that evaluated the previous (<60 years old) or current occupational level, six obtained statistically significant results [14,28,34,45,46,53] and six did not [30,35,36,37,44,51] (Appendix A). Of the studies with statistical significance, four showed that people with low current occupational levels or who were retired or unemployed had a higher percentage of malnutrition or malnutrition risk [14,28,34,45]. Two of these evaluated the OR [34,45] and one of them the β coefficient [34]. One study established that unemployed people had 22.2 times more risk of malnutrition [45]. In another study, the OR and the β coefficient revealed that administrative professions protect against malnutrition [34]. The OR of the other significant study showed that working (<60 years old or currently) as a farmer or animal breeder increased the risk of malnutrition [46]. Only one study obtained significant opposite results, expressing a high percentage of malnutrition among working elderly people [53].

### 3.7. Feeling of Loneliness and Malnutrition/Malnutrition Risk

Of the total six studies that evaluated the feeling of loneliness, four were related significantly with malnutrition or malnutrition risk [13,17,21,33], one suggested a significant trend [34], and only one did not show significant results [36] (Appendix A). Of the significant studies, two resolved that the elderly with a greater feeling of loneliness had a higher prevalence of malnutrition [17,33]. Of these studies, one evaluated the HR [33], showing an increased risk of malnutrition of 1.47 in the elderly who had feelings of loneliness. In contrast, two studies had significant opposite results, showing that the elderly with a lower score in the MNA questionnaire (malnutrition) did not express feelings of loneliness [13,21].

### 3.8. Place of Residence and Malnutrition/Malnutrition Risk

A total of three studies evaluated the place of residence, which was differentiated between urban and rural. Only one study established significant relationships with malnutrition or malnutrition risk [28] and two did not show significant results [39,51] (Appendix A). The only study with significant results showed that the elderly who lived in rural areas had worse nutritional status and more malnutrition or malnutrition risk, in comparison with those who lived in urban areas [28].

### 3.9. Food Expenditure and Malnutrition/Malnutrition Risk

Only one study evaluated the relation between food expenditure and malnutrition or malnutrition risk, and it did not show statistically significant results [41] (Appendix A).

### 3.10. Quality of the Articles Included in the Systematic Review

Of the total 40 studies included, 16 were of high quality [17,22,24,25,26,29,30,33,38,41,43,45,46,48,51,53], 23 of medium quality [13,14,16,18,19,20,21,23,27,31,32,34,35,36,37,39,40,42,44,47,49,50,52], and one of low quality [28]. Of the 16 studies of high quality, one of them was a cohort study [33]. Of the 23 studies of medium quality, three of them were cohort studies [16,19,50]. Regarding the questionnaire, questions 1, 2, 9, 11, and 14, which refer to the research question, study population, exposure measures, and assessment, outcome measures, and statistical analyses, respectively, were the most reported by the studies. In contrast, questions 5 and 10, which refer to the sample size justification and the repeated exposure assessment, respectively, were the least reported by the studies. The quality of the studies is shown in Table 1 and Appendix A.

### 3.11. Meta-Analysis

Of the total of 40 studies included in the systematic review, 16 cross-sectional studies were included in the meta-analysis [13,20,21,25,26,29,30,35,36,39,42,43,45,51,52,53]. The meta-analysis included a sample of 34,703 individuals, of which 10,755 people had or were at risk of malnutrition and 23,948 had good nutritional status; the Forest Plot is shown in Figure 2. In general, evaluating all variables at once, it was observed that having a low educational level, living alone, being single, widowed, or divorced, and having a low income level increased malnutrition and the risk of malnutrition in the elderly (OR: 1.83; 95% Confidence Interval (CI): 1.73–1.93; *p* < 0.001; I^2^ = 94%; *p* about heterogeneity < 0.001) (Figure 2). The 16 cross-sectional studies included in the meta-analysis were analyzed by subgroups depending on the socioeconomic variable [13,20,21,25,26,29,30,35,36,39,42,43,45,51,52,53].

#### 3.11.1. Educational Level and Malnutrition/Malnutrition Risk

Thirteen studies that evaluated the educational level in relation to malnutrition or malnutrition risk were included [20,25,26,29,30,35,36,39,43,45,51,52,53]. This meta-analysis included a sample of 10,734 individuals, of which 3233 had or were at risk of malnutrition. It was observed that having a low educational level is a risk factor for malnutrition (OR: 1.48; 95% CI: 1.33–1.64; *p* < 0.001; I^2^ = 91%; *p* about heterogeneity < 0.001) (Figure 2).

#### 3.11.2. Living Alone or Cohabiting and Malnutrition/Malnutrition Risk

Ten studies that evaluated the relationship between living alone and malnutrition or malnutrition risk were included [21,25,26,36,39,42,45,51,52,53]. This meta-analysis included a sample of 9042 individuals, of which 3034 had or were at risk of malnutrition. It was observed that living alone is a risk factor for malnutrition (OR: 1.92; 95% CI: 1.73–2.14; *p* < 0.001; I^2^ = 90%; *p* about heterogeneity < 0.001) (Figure 2).

#### 3.11.3. Marital Status and Malnutrition/Malnutrition Risk

Twelve studies that evaluated the marital status and malnutrition or malnutrition risk were included [13,26,29,30,35,36,39,42,45,51,52,53]. This meta-analysis included a sample of 10,083 individuals, of which 2843 had or were at risk of malnutrition. It was observed that being single, widowed, or divorced (in comparison to being married) is a risk factor for malnutrition (OR: 1.73; 95% CI: 1.57–1.90; *p* < 0.001; I^2^ = 91%; *p* about heterogeneity < 0.001) (Figure 2).

#### 3.11.4. Income Level and Malnutrition/Malnutrition Risk

Five studies that evaluated the income level and malnutrition or malnutrition risk were included [20,25,35,43,51]. This meta-analysis included a sample of 4844 individuals, of which 1645 had or were at risk of malnutrition. It was observed that having a low income level is a risk factor for malnutrition (OR: 2.69; 95% CI: 2.35–3.08; *p* < 0.001; I^2^ = 89%; *p* about heterogeneity < 0.001) (Figure 2).

## 4. Discussion

The present systematic review and meta-analysis of observational studies, which were mainly cross-sectional, supports the hypothesis that social and economic factors are related to malnutrition and malnutrition risk in people aged ≥ 60 years. Focused on the meta-analysis outcomes, according to the socioeconomic factor, in a descendant order, low income level, living alone, being single, widowed, or divorced, and low educational level had a greater relationship with malnutrition and malnutrition risk in the elderly. Meanwhile, there were not enough articles providing scientific evidence to make a meta-analysis of some of the other socioeconomic factors, such as low occupational level (e.g., farmer, breeder, housewife, laborer, or hard manual or physical work), feeling of loneliness, living in rural areas, and food expenditure. Thus, it was not possible to prove whether these factors act as a malnutrition risk factor in the elderly.

In the current meta-analysis, it was observed that low educational level is a risk factor for malnutrition, suggesting the predisposition to malnutrition. A possible explanation is that individuals with a higher educational level have more knowledge about nutrition and health [55], leading to a healthier and more varied diet and, therefore, a better nutritional status than individuals with a low educational level.

Another identified risk factor from the results of the present meta-analysis is living alone, which showed a significant association with the risk of developing malnutrition. The elderly who lived alone had a greater malnutrition risk compared to others who cohabit with someone—either family, acquaintances, or friends. Three literary reviews [56,57,58] obtained the same results as the present review and related the fact of living alone with malnutrition. Moreover, gender can influence the risk of malnutrition. A review of the literature observed that men who lived with their wives had better nutrition than men who lived alone, because their wives were in charge of regularly cooking and grocery shopping [59]. Another explanation is that the elderly who lived alone had a worse nutritional status due to the lack of social interaction with other people [55]. More specifically, in a study investigating the effect of the presence of others, both within the household and during meals, on caloric intake in homebound older adults, it was concluded that the elderly who ate with other people consumed more calories than those who ate alone [60]. Thus, a simple and inexpensive way to increase caloric intake is to make arrangements for family members or caregivers to eat with them [60]. In contrast to the results obtained in the present meta-analysis, in two systematic reviews of the determinants of malnutrition, living alone was not identified as a risk factor for malnutrition [4,61].

In reference to marital status, the present meta-analysis established an association between being single, widowed, and divorced and the risk of malnutrition. In the same way to the present results, one review concluded that widowed individuals, independently of gender, were more vulnerable to the risk of nutritional deficiencies [62]. The possible explanations could be that, firstly, widowhood is associated with poor eating habits and with less enjoyment of eating [63], for example, the loss of the social interaction culturally related to food. Secondly, the wife’s death implies a deterioration in the husband’s nutritional status as a consequence of inexperience in food tasks in this generation [59]. Despite the evidence shown so far, one systematic review established that the death of a spouse was not associated with malnutrition in the elderly [4].

Furthermore, the present meta-analysis also established an association between lowest income levels and a higher risk of developing malnutrition in the elderly. One explanation is based on the fact that the healthiest or freshest food is the most expensive, so are not accessible for the elderly with few economic resources [64]. Thus, all of these low income-associated aspects could increase the risk of malnutrition and could be a determinant of malnutrition in the elderly.

Another socioeconomic factor evaluated is the low previous (<60 years old) or current occupational level, namely, farmer, breeder, housewife, laborer, hard manual or physical work, or unemployment, which were identified, in the present systematic review, as a malnutrition risk factor. However, there were not enough articles to perform a meta-analysis. A possible explanation is that those elderly with a low occupational level usually have a lower economic level too, so they might not have access to high-quality nutritional food.

Regarding the feeling of loneliness, which is separate from social isolation, the present systematic review obtained low numbers of articles relative to the risk of malnutrition. Only two cross-sectional studies showed that the elderly with a greater feeling of loneliness were at greater risk of malnutrition [17,33], while other reviews did not find this association [57,62,65,66]. One review stated that the aging process was related to psychosocial and environmental changes, such as loneliness [65]. These changes could have a negative effect on nutritional status [65]. Probably, the feeling of loneliness in the elderly causes a loss of appetite and interest in meals. The feeling of loneliness, along with isolation and decreased social interactions, worsens the nutritional vulnerability in the elderly [62]. On the contrary, one systematic review of prospective studies suggested that the feeling of loneliness was not related to the risk of malnutrition [61].

The present meta-analysis could not establish a reliable association regarding the place of residence, either urban or rural, and malnutrition risk in the elderly. Of all of the studies included in the review, only one cross-sectional study showed that people in Iran who lived in rural areas had a greater risk of malnutrition [28]. In the same way, a cross-sectional study concluded that the MNA score was significantly lower in people who lived in rural areas compared to others who lived in urban areas, while the rural elderly had lower educational and income levels than the urban elderly [67]. As mentioned earlier, low education and low income levels could negatively affect their nutritional status [67]. Finally, another socioeconomic factor that could not establish a reliable association with malnutrition in this systematic review and meta-analysis was that of food expenditure, probably related to the economic level of the elderly. As stated above, a low economic level means not having enough money for food [68].

This systematic review and meta-analysis presents different limitations. First of all, the present systematic review was about observational studies, mainly cross-sectionals. Future studies should evaluate the influence of socioeconomic factors in malnutrition of randomized controlled intervention studies to obtain maximum evidence [69]. The second limitation was that not all studies used a unified definition to evaluate malnutrition or malnutrition risk. For example, from the studies included in the meta-analysis, seven evaluated malnutrition [20,29,39,42,43,52,53], two evaluated only the risk of malnutrition [35,36], and seven evaluated malnutrition and risk of malnutrition at the same time [13,21,25,26,30,45,51]. It should be clarified in future studies the distinction between malnutrition and risk of malnutrition to improve the quality of the evidence. In addition, studies either did not use the same tool to evaluate socioeconomic factors, or the tool used was not described appropriately. Thirdly, few studies evaluated the feeling of loneliness, the place of residence, or the food expenditure; therefore, no meta-analysis could be done on how these socioeconomic factors were associated with malnutrition and malnutrition risk. In addition, specifically on the occupational level, the jobs analyzed did not represent all professions. Fourthly, socioeconomic psychosocial factors, such as loneliness, isolation, or depression need to be studied, since they have an important burden in the development of malnutrition in the elderly [62]. The last limitation is that when performing the meta-analysis, a high level of heterogeneity of the included studies was observed, which represents an important limitation [70]. This heterogeneity could be due to the inclusion of the general population (illness and not illness individuals), as it excluded studies that only included ill people.

All of the results of the present systematic review and meta-analysis serve to better understand the social and economic risk factors associated with malnutrition in the elderly ≥60 years old. Moreover, the identification of social and economic risk factors allows us to avoid them, and thus to reduce or prevent malnutrition or/and malnutrition risk and to maintain a good nutritional status of the elderly. Thus, regarding the evidence provided in the present systematic review and meta-analysis, the following table shows a proposal of actions on socioeconomic risk factors to reduce or prevent malnutrition and malnutrition risk. These actions can provide a basis for future nutritional interventions on socioeconomic factors to diminish malnutrition in the elderly. However, more research is needed to confirm that these interventions are effective in combating each socioeconomic factor (Table 2).

## 5. Conclusions

Malnutrition and malnutrition risk could be reduced in the elderly by increasing their economic level, supporting those living alone or who are single, widowed, or divorced, and improving lifelong learning.

## Figures and Tables

**Figure 1 nutrients-12-00737-f001:**
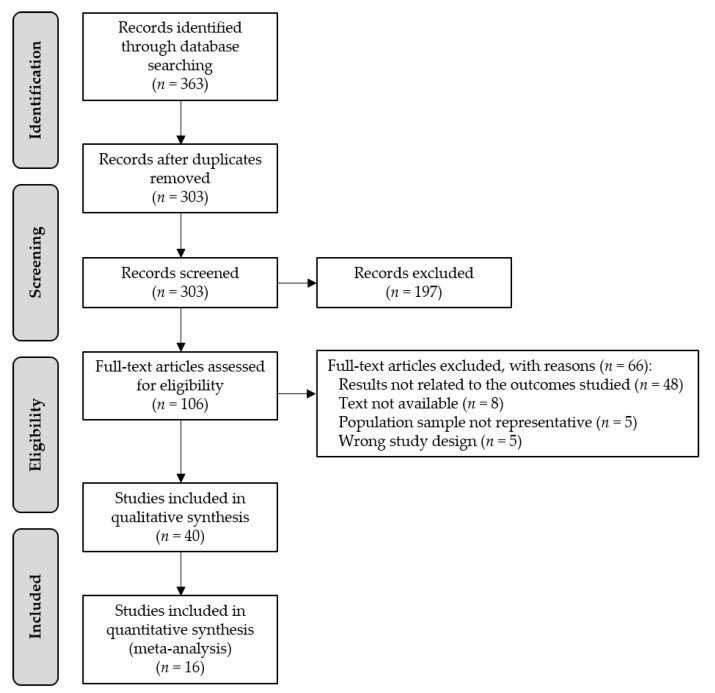
Preferred Reporting Items for Systematic Reviews and Meta-Analyses (PRISMA) flow diagram of the study selection.

**Figure 2 nutrients-12-00737-f002:**
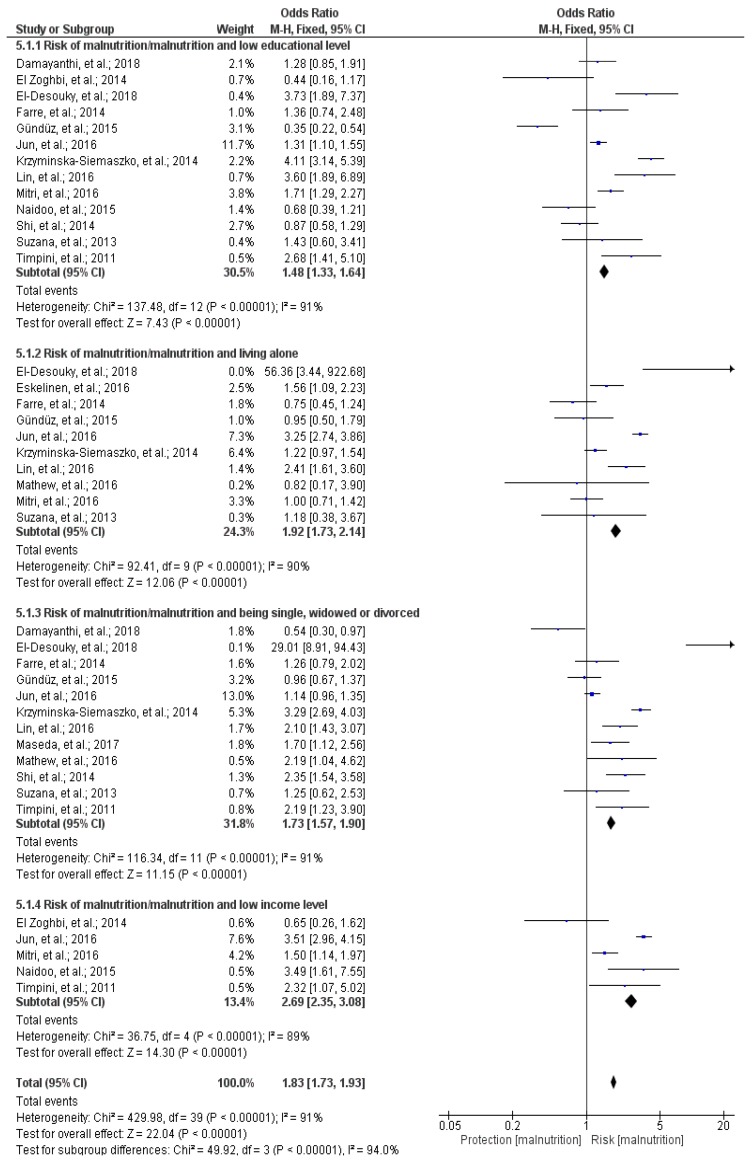
Forest plot of the relationship between malnutrition and/or malnutrition risk and socioeconomic factors (low educational level, living alone, marital status of single, widowed, or divorced, and low income level) individually and by subgroups, in alphabetical order, in the elderly (aged ≥ 60 years).

**Table 1 nutrients-12-00737-t001:** Quality assessment and socioeconomic variables analyzed of the observational studies included in the systematic review.

	Quality	Socioeconomic Variables Analyzed
Bardon et al., 2018 [19]	Medium	Educational level, living alone, marital status
Boulos et al., 2016 [17]	High	Feeling of loneliness
Cabrera et al., 2007 [18]	Medium	Educational level
Chen et al., 2007 [16]	Medium	Educational level, marital status, income level
Damayanthi et al., 2018 [29]	High	Educational level, living alone, marital status, income level
Debnath et al., 2017 [44]	Medium	Educational level, living alone, marital status, income level, occupational level
Donini et al., 2013 [14]	Medium	Educational level, living alone, marital status, income level, occupational level
El Zoghbi et al., 2014 [20]	Medium	Educational level, income level
El-Desouky et al., 2018 [45]	High	Educational level, living alone, marital status, income level, occupational level
Eskelinen et al., 2016 [21]	Medium	Living alone, feeling of loneliness
Farre et al., 2014 [26]	High	Educational level, living alone, marital status
Ferdous et al., 2009 [41]	High	Educational level, marital status, income level, food expenditure
Ferra et al., 2012 [22]	High	Educational level, living alone, income level
Fjell et al., 2018 [23]	Medium	Marital status
Gündüz et al., 2015 [52]	Medium	Educational level, living alone, marital status
Jésus et al., 2017 [46]	High	Educational level, marital status, occupational level
Ji et al., 2012 [37]	Medium	Educational level, living alone, income level, occupational level
Johansson et al., 2009 [40]	Medium	Living alone
Jun et al., 2016 [51]	High	Educational level, living alone, marital status, income level, occupational level, place of residence
Krzymińska-Siemaszko et al., 2014 [39]	Medium	Educational level, living alone, marital status, income level, place of residence
Lengyel et al., 2017 [50]	Medium	Living alone
Lin et al., 2016 [53]	High	Educational level, living alone, marital status, occupational level
Madeira et al., 2018 [24]	High	Educational level, marital status, income level
Maseda et al., 2017 [13]	Medium	Educational level, living alone, marital status, feeling of loneliness
Mathew et al., 2017 [42]	Medium	Educational level, living alone, marital status, income level
Mitri et al., 2016 [25]	High	Educational level, living alone, income level
Mokhber et al., 2011 [28]	Low	Educational level, living alone, occupational level, place of residence
Naidoo et al., 2015 [43]	High	Educational level, living alone, income level
Olayiwola et al., 2006 [47]	Medium	Educational level, income level
Park et al., 2014 [27]	Medium	Educational level, living alone, marital status, income level
Ramage-Morin et al., 2013 [49]	Medium	Educational level, living alone
Rodriguez-Tadeo et al., 2011 [32]	Medium	Living alone
Romero-Ortuno et al., 2010 [31]	Medium	Living alone
Schilp et al., 2011 [33]	High	Educational level, marital status, income level, loneliness feeling
Shi et al., 2014 [30]	High	Educational level, marital status, income level, occupational level
Söderhamn et al., 2012 [34]	Medium	Marital status, occupational level, feeling of loneliness
Suzana et al., 2013 [36]	Medium	Educational level, living alone, marital status, occupational level, feeling of loneliness
Timpini et al., 2011 [35]	Medium	Educational level, marital status, income level, occupational level
Westergren et al., 2014 [38]	High	Living alone
Wham et al., 2015 [48]	High	Educational level, living alone, marital status

**Table 2 nutrients-12-00737-t002:** Proposal of nutritional interventions to prevent malnutrition and malnutrition risk based on the socioeconomic factors presented in elderly.

Socioeconomic Factors	Nutritional Interventions
Low educational level	Create promotion and education campaigns for healthy eating [71]
	Form information meetings on nutrition for patients and their families [72]
Living alone Feeling of loneliness Single, widowed, divorced	Socialize during meals, avoid eating alone; go to the relative’s home [72]
	Do group activities and share meals [72]
	Cook a greater quantity of food and keep it in the fridge for another day of the week
Low income level Low occupational level Low food expenditure	Buy healthy white label products
	Buy seasonal products
	Make a shopping list to avoid buying unnecessary things
	Make a weekly menu and adjust the grocery shop to that menu
	Buy basic necessities, avoid superfluous products
	Resort to social organizations or social programs when it is necessary [71,72]
Rural place of residence	Buy local and proximity products
	Cook traditional local recipes
	Inform the elderly about healthy eating in local shops

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
