# Peer review of "Social and Economic Factors and Malnutrition or the Risk of Malnutrition in the Elderly: A Systematic Review and Meta-Analysis of Observational Studies"

_nutrients, 2020, doi:10.3390/nu12030737_

Round 1
Reviewer 1 Report
Need to define "malnutrition" and "at risk of malnutrition / risk of malnutrition". Authors can use ESPEN guidlines for this purpose. Better to remove the word condition from malnutrition condition all over the paper. Keep only malnutrition which is solved when they define malnutrition.
Line 3 - How far the word incidence is appropriate? usually incidence defines number of new cases
Line 68 - What do you mean by low? It was explained in the discussion. Authors have to define low occupational level in the method which will give more clarity.
Line 93 - studies were included over 60 years, but the objective - line 73- indicate 60 and above. How do you explain it. It shows from the beginning studies with 60 years are excluded in this systematic review, which is a wrong selection.
Line 97 - excluding illness and co-morbidities create bias. It is a major confounder. This particular age group main cause of malnutrition is illness or co-morbidities. Majority of this age group are with some illness. When authors remove it, this will really restrict many studies getting selected. Need a well explanation for this action, even in the discussion.
line 127 - need correction of grammer
line 322 - studies > 60 years. need the explanation.
line 326 and 361- need to clarify low level as mention above. not clear
line 367-369 - need to shift to method
line 409 - correct to >=60 years
Author Response
Comments and Suggestions for Authors
Thank you for the comments of the reviewer.
Question 1. Need to define "malnutrition" and "at risk of malnutrition / risk of malnutrition". Authors can use ESPEN guidelines for this purpose. Better to remove the word condition from malnutrition condition all over the paper. Keep only malnutrition, which is solved when they define malnutrition.
As the reviewer commented, we deleted the word conditions in the manuscript and Supplementary Tables.
Moreover, we included the definition of malnutrition and malnutrition risk in Introduction section (Pg 1-2, ln 38-43; 48-53): “From the European Society for Clinical Nutrition and Metabolism (ESPEN) definition, malnutrition is diagnosed by a body mass index (BMI) of <18.5 kg/m2 or by meeting two of these three criteria: unintentional weight loss (>10% in an indefinite time period or >5% over the last 3 months) combined with either a low BMI (BMI of <20 kg/m2 if <70 years of age, or <22 kg/m2 if ≥70 years of age) or a low fat-free mass index (FFMI) score (FFMI of <15 and <17 kg/m2 in women and men, respectively) [7]. However, before reaching the need of a malnutrition diagnosis, the use of validated tools could be used to determine malnutrition risk [6,7]. According to the ESPEN, the Mini Nutritional Assessment (MNA) is the most effective tool for screening and evaluating the risk of malnutrition in the elderly [6,10,11], composed of four areas related to anthropometry, clinical status (illness, medications, psychological stress, neuropsychological problems), dietary assessment, and self-perception about health and nutrition [10]”
Malnutrition risk could be not deleted from the manuscript because a huge part of the articles included, reference risk of malnutrition instead of malnutrition, or combined the two definitions together. We included in the limitation section (Pg 12, ln 408-415): “The second limitation was that not all studies used a unified definition to evaluate malnutrition or malnutrition risk. For example, from the studies included in the meta-analysis, seven evaluated malnutrition [20,29,39,42,43,52,53], two evaluated only the risk of malnutrition [35,36], and seven evaluated malnutrition and risk of malnutrition at the same time [13,21,25,26,30,45,51]. It should be clarified in future studies the distinction between malnutrition and risk of malnutrition to improve the quality of the evidence. In addition, studies either did not use the same tool to evaluate socioeconomic factors, or the tool used was not described appropriately.”
Question 2. Line 3 - How far the word incidence is appropriate? Usually incidence defines number of new cases.
We deleted the incidence information, due to it is not necessary in this case.
Question 3. Line 68 - What do you mean by low? It was explained in the discussion. Authors have to define low occupational level in the method, which will give more clarity.
As the reviewer requested, we introduced in the methods the explanation about “low” (Pg 3, ln 121-124): “The previous or current occupational level was divided into low occupational levels (such as farmer, breeder, housewife, laborer, or hard manual or physical work) and high occupational levels (such as employee, businessman, worker, administration, or jobs involving higher intellectual effort).”
Question 4. Line 93 - studies were included over 60 years, but the objective - line 73- indicate 60 and above. How do you explain it? It shows from the beginning studies with 60 years are excluded in this systematic review, which is a wrong selection.
There was an error in the study selection explanation. We selected studies with a population of ≥60 years old. We changed this in the manuscript, and we corrected this in all the manuscript.
Question 5. Line 97 - excluding illness and co-morbidities create bias. It is a major confounder. This particular age group main cause of malnutrition is illness or co-morbidities. Majority of this age group are with some illness. When authors remove it, this will really restrict many studies getting selected. Need a well explanation for this action, even in the discussion.
We considered including studies of the general population. These studies could include individuals with illness or not. However, we considered excluding the studies that only include illness individuals, due to be able to generalize the results obtained in the present meta-analysis.
We introduced in the Methods section (Pg 3, ln 104-107): “Additionally, studies which only included populations with illness or comorbidities were also excluded in order to focus on the general population, who was not selected taking into account a specific disease, to be able to generalize the results.”
We introduced in the limitation section (Pg 12, ln 422-424): “This heterogeneity could be due to the inclusion of the general population (illness and not illness individuals), as it excluded studies that only included ill people.”
Question 6. line 127 - need correction of grammar
We send this manuscript to English and grammar correction, to improve the grammar flaws.
Question 7. line 322 - studies > 60 years. Need the explanation.
This was a mistake, we improved it in the manuscript (Pg 11, ln 334-336): “The present systematic review and meta-analysis of observational studies, which were mainly cross-sectional, supports the hypothesis that social and economic factors are related to malnutrition and malnutrition risk in people aged ≥60 years.”
Question 8. line 326 and 361- need to clarify low level as mention above. Not clear.
We introduced little explanation about low level of occupational activity in Discussion section (Pg 11, ln 339-343; 373-380): “Meanwhile, there were not enough articles providing scientific evidence to make a meta-analysis of some of the other socioeconomic factors, such as low occupational level (e.g., farmer, breeder, housewife, laborer, or hard manual or physical work), feeling of loneliness, living in rural areas, and food expenditure.”; and, “Furthermore, the present meta-analysis also established an association between lowest income levels and higher risk of developing malnutrition in the elderly. One explanation is based on the fact that the healthiest or freshest food is the most expensive, so are not accessible for the elderly with few economic resources [64]. Thus, all of these low income-associated aspects could increase the risk of malnutrition and could be a determinant of malnutrition in the elderly.
Another socioeconomic factor evaluated is the low previous (<60 years old) or current occupational level, namely, farmer, breeder, housewife, laborer, hard manual or physical work, or unemployment, which were identified, in the present systematic review, as a malnutrition risk factor.”
Question 9. line 367-369 - need to shift to method
We put this explanation in the Method section (Pg 3, ln 121-124): “The previous or current occupational level was divided into low occupational levels (such as farmer, breeder, housewife, laborer, or hard manual or physical work) and high occupational levels (such as employee, businessman, worker, administration, or jobs involving higher intellectual effort).”
Question 10. line 409 - correct to >=60 years
We corrected this from the manuscript (pg 12, ln 425-426): “All of the results of the present systematic review and meta-analysis serve to better understand the social and economic risk factors associated with malnutrition in the elderly ≥60 years old.”
Reviewer 2 Report
Dear authors,
Thank you very much for the brilliant work you have done with the review and meta-analysis.
If you want to, you could add the reason why you chose pubmed und SCOPUS as electronic databases and not any other.
Author Response
Comments and Suggestions for Authors:
Dear authors, Thank you very much for the brilliant work you have done with the review and meta-analysis.
Thank you for the consideration of our work, we really appreciate your comments.
Question 1. If you want to, you could add the reason why you chose PubMed und SCOPUS as electronic databases and not any other.
As the reviewer suggested, we included in the Methodology section (Pg 2, ln 87-90): “An exhaustive literature search in the PubMed and SCOPUS electronic databases was carried out, as these databases are considered the largest and the most multidisciplinary, covering medical and social sciences journals that include the principal outcomes of the present meta-analysis (malnutrition and socioeconomic factors).”